# Solitary Fibrous Tumors of the Lung: A Clinicopathological Analysis of 52 Cases

Ying-Xia Wang [1], Yan Zhong [2], Su-Su Fan [2], Yu-Shan Zhu [2], Xue-Rong Peng [2] and Xuan Zhang [2,*]

1  Department of Pathology, The First Affiliated Hospital of Kunming Medical University, Kunming 650032, China
2  Yunnan Key Laboratory of Pharmacology for Natural Products, School of Pharmaceutical Sciences, Kunming Medical University, Kunming 650500, China
*  Correspondence: zhangxuan@kmmu.edu.cn

**Abstract:** Objective: To explore the clinicopathological features of solitary fibrous tumors (SFTs) of the lung. Methods: We collected the clinical data of 52 patients with SFTs of the lung confirmed by pathology, and summarized the clinical, radiological, and morphological features, the immunophenotypes, and the prognosis of SFTs. Results: Fifty-two cases of SFTs of the lung were enrolled in this study, including 51 cases of borderline and 1 case of malignancy, 22 males and 30 females. The average onset age was 52.7 years. The lower lobe of the left lung was the preferred site of SFTs, accounting for 30.77% (16/52). Chest CT showed regular and well-demarcated soft tissue density mass, and the tumor size of most cases (46/52, 88.46%) was 1–10 cm. Morphological features: The distribution of tumor cells showed sparse and dense areas. Tumor cells were spindle-shaped, in whorls or hemangiopericytoma-like conformation. Atypia, mitotic figures, and necrosis were found. Immunohistochemistry showed positive expression of CD34, STAT6, Vimentin, BCL2, and CD99. Ki-67 was ≤ 5% in borderline SFTs and >20% in a malignant SFT. Conclusions: Solitary fibrous tumors of the lung occur more frequently in middle-aged and elderly people, and there is no significant difference in gender. The lower lobe of the left lung is the preferred site of SFTs. The size of most SFTs is 1–10 cm. Chest CT shows morphologically regular and well-demarcated soft tissue density mass. Pathologically, SFTs of the lung are mostly borderline and occasionally malignant. Immunohistochemistry shows the positive expression of CD34, STAT6, Vimentin, BCL2, and CD99.

**Keywords:** solitary fibrous tumors; lung; pathological features; immunohistochemistry; malignancy; borderline





## 1. Introduction

Solitary fibrous tumors (SFTs) are uncommon spindle cell tumors of mesenchymal origin [1]. SFTs can occur in all organs of the body, including the nasopharynx, bladder, prostate, limbs (especially thighs), back, trunk, head and neck, pleura, peritoneum, pelvis, meninges, thyroid, breast, and pharynx [2]. SFTs can be found by chest CT, which shows the morphology of well-demarcated soft tissue density mass, but CT lacks a specific diagnostic basis [3]. SFTs are mostly borderline tumors with an ICD-O code of 1, whereas some SFTs are malignant with an ICD-O code of 3. Most SFTs have a good prognosis after surgical resection. A few larger tumors may be malignant. Both borderline and malignant SFTs may relapse, but the recurrence rate of malignant SFTs is higher [4].

SFTs are known for having an indolent course, with relatively infrequent metastasis. SFTs metastasize in 5–25% of cases; it has historically been challenging to determine which specific tumor and patient characteristics predict aggressive behavior. Demicco reported a four-variable model to predict the metastatic risk of SFTs [5]. This risk-stratification model is one of the most accepted risk-stratification criteria for SFTs. Most SFTs occurring in the chest originate from the pleura, rarely do SFTs originate from the lung. As SFTs rarely occur in the lung, the current reports of SFTs of the lung were fewer than 10 cases. In this

study, data from 52 cases of SFTs occurring in the lung were collected, and a comprehensive analysis was performed to learn the clinical, radiological, and morphological features, immunophenotypes, and prognosis of SFTs of the lung.

## 2. Methods

### 2.1. Case Collection

The data from 52 patients with confirmed SFTs occurring in the lung and treated in the First Affiliated Hospital of Kunming Medical University from January 2009 to October 2021 were collected. Prior to the inclusion of the cases, two pathologists with senior professional titles were asked to review the pathological sections and confirm the correct diagnosis. The medical records of the patients included in the study were reviewed to understand the chief complaint, history of the disease, surgical findings, and tumor size. The chest CT findings were reviewed to understand the CT manifestations and characteristics of all SFTs.

### 2.2. Sample Processing

After sampling, the specimens were fixed with 4% neutral formaldehyde for 4–16 h, dehydrated in 75% alcohol, 85% alcohol, 95% alcohol, 95% alcohol, anhydrous ethanol, anhydrous ethanol, xylene and xylene, successively, then immersed in paraffin wax with a melting point of 56–58 °C in two cylinders, successively. Then, the specimens were embedded in paraffin with a melting point of 62–64 °C to make wax pieces. Each piece was cut into 4 μm sections. Multiple 4 μm sections were cut from typical parts of the tumor. After roasting, dewaxing, debenzening, hematoxylin-eosin staining, dehydration, transparency, and sealing, HE sections were made. Immunohistochemical antibodies such as CD34, STAT6, Bcl-2, CD99, Vimentin, PR, ER, and Ki-67 were selected for immunohistochemical staining using an en-Vision two-step method by routine dewaxing and debenzenization, hydration, heat repair in pressure cookers, washing with PBS buffer solution, primary antibody incubation at 37 °C for 1 h, washing with PBS buffer solution 3 times, secondary antibody incubation at room temperature for 20 min, washing with PBS buffer solution 3 times, DAB color development for 1 to 5 min; then, after hematoxylin lining staining with a nucleus, the dehydrated, transparent and sealed, immunohistochemical sections were prepared.

### 2.3. Metastatic Risk Assessment

Metastatic risk scores were calculated using a four-variable model reported by Demicco in 2017 [5]. Patient age was scored as 0 if <5 years and 1 if ≥55 years. Tumor size was scored as 0 if <5 cm, 1 if 5 to <10 cm, 2 if 10 to 15 cm, or 3 if ≥15 cm. Mitotic activity was scored as 0 if <1 mitotic figure/10 HPF, 1 if 1–3 mitotic figures/10 HPF, or 2 if ≥4/10 HPF. Tumor necrosis was scored 0 if <10%, 1 if ≥10%. Total scores were summed, and scores of 0–3 were considered low risk, 4–5 as intermediate risk, and 6–7 as high risk.

## 3. Results

### 3.1. Clinical Features

The onset age of SFTs of the lung in this study was 25–86 years old, with an average age of 52.7 years and a median age of 52 years. The onset age showed normal distribution, and most cases of SFTs occurred at the age of 41–60 years, accounting for 61.5% (32/52) of the total cases (Figure 1).

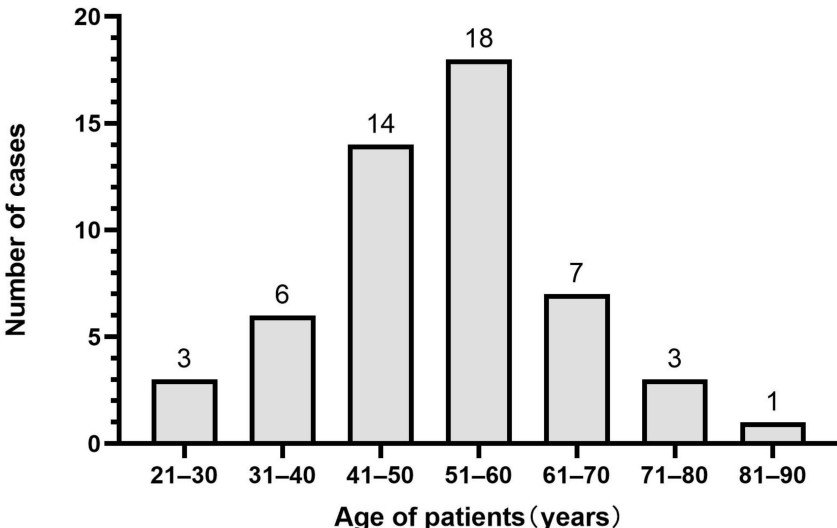

**Figure 1.** The onset age distribution of solitary fibrous tumors (SFTs) of the lung.

The enrolled patients with SFTs of the lung included 22 males and 30 females, no significant difference in gender was found. SFTs occurred in the lower lobe of left lung in 16 cases, accounting for 30.77%, the upper lobe of left lung in 11 cases, accounting for 21.15%, the lower lobe of right lung in 11 cases, accounting for 21.15%, the upper lobe of right lung in 8 cases, accounting for 15.38%, the middle lobe of right lung in 5 cases, accounting for 9.62%, and the upper and lower lobes of left lung in 1 case, accounting for 1.92% (Figure 2). Among the 52 patients with SFTs of the lung, 18 (34.6%) patients had no obvious symptoms, and SFTs were accidentally found by chest CT scanning for lung cancer screening during physical examination; 34 (65.4%) patients had symptoms such as cough, less phlegm, chest tightness, chest pain and shortness of breath. One of the patients had concurrent squamous cell carcinoma, one patient had concurrent microinvasive adenocarcinoma, and the remaining 50 patients all had SFTs alone, including 1 case of malignant SFTs.

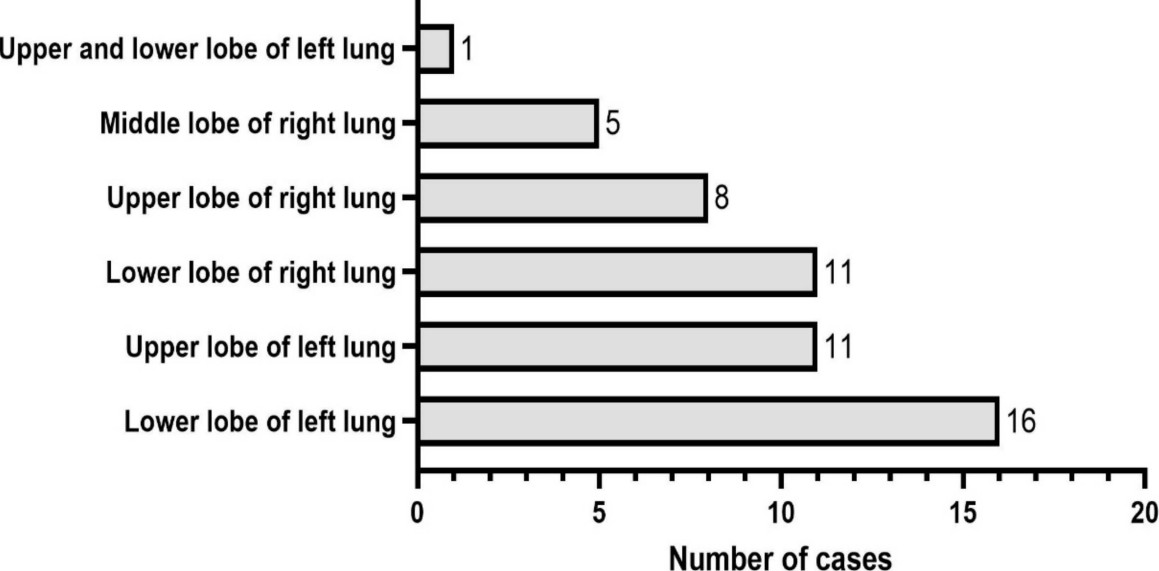

**Figure 2.** The occurrence site distribution of solitary fibrous tumors (SFTs) of the lung.

*3.2. Imaging Features*

Some of the patients underwent full staging CT scan including abdomen and pelvis as part of the original diagnosis, but no abnormality was found in abdomen, pelvis or other parts except the lung.

The typical chest CT findings showed a regular and well-demarcated soft tissue density mass, which could be nodular (Figure 3, red arrow), and the surrounding lung tissue and bronchi were compressed and narrowed. SFTs occurred most frequently in the hilar area or near the visceral pleura. The maximum diameter of the mass varied from 0.7 cm to 23 cm, with a larger SFT (5.0 × 4.0 × 4.0 cm) of one case occupying two lobes of the lung. Most cases of SFTs were in the range of 1–10 cm, accounting for 88.46% (46/52), only two cases of SFTs were less than 1 cm, and four cases of SFTs were over 10 cm.

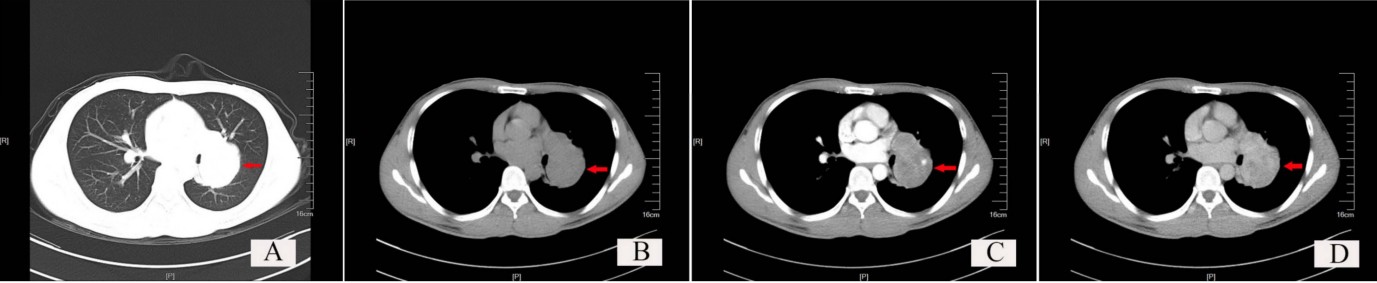

**Figure 3.** Chest CT findings of a typical solitary fibrous tumor (SFT) of the lung. (**A**) Lung window: Red arrow shows a well-demarcated soft tissue density shadow, which is nodular and located at the hilum, with regular morphology. (**B**) Plain scan of the mediastinal window: the red arrow indicates the tumor. (**C**) Arterial phase scan of the mediastinal window: red arrow indicates the tumor. (**D**) Venous phase scan of the mediastinal window: red arrow indicates the tumor.

### 3.3. Morphological Features

The distribution of the tumor cells showed sparse and dense areas (Figure 4A); some tumor cells showed in whorls (Figure 4B) or hemangiopericytoma-like conformation (Figure 4E); a few tumor cells were distributed in beams or papillary, with numerous glass-like collagen fibers interspersed between the tumor cells (Figure 4D,F). The tumor cells were spindle-shaped, and relatively uniform in size (Figure 4C), with or without significant atypia. The tumor cells in the dense areas showed epithelioid cell morphology, and mitotic figures could be observed. Large irregular branched or/and antler-like blood vessels were observed in the tumors (Figures 4A and 5A), and often showed hyaline degeneration, with tumor cells distributed around the blood vessels. In this study, there was only one case of malignant SFTs among the 52 cases of SFTs. In the case of malignant SFT, the spindle-shaped tumor cells were densely packed, no significant density partition was observed (Figure 5A,B), the tumor cells had obvious atypia (Figure 5C), multiple mitotic figures (more than 4 mitotic figures per 10 high-power fields) were observed (Figure 5D, red arrow), usually more than 4 mitotic figures per 10 high-power fields, and tumor necrosis was observed (Figure 5E).

### 3.4. Immunophenotypes

Immunohistochemistry results (Figures 4G–L and 5F–I) revealed 100% positivity for Vimentin (52/52) and STAT-6 (40/40), 94.23% positivity for CD34 (49/52), 85.71% positivity for CD99(42/49), and 78.43% positivity for BCL-2 (40/51). Some cases of SFTs showed positivity for ER (2/3) and PR (6/9) (Table 1). The Ki-67 index is an important indicator to distinguish benign and malignant tumors. In 52 cases of SFTs of the lung, the Ki-67 indexes of 51 cases of borderline SFTs were <5%, accounting for 98.07%; the Ki-67 index of only one case of malignant SFT was >20%.

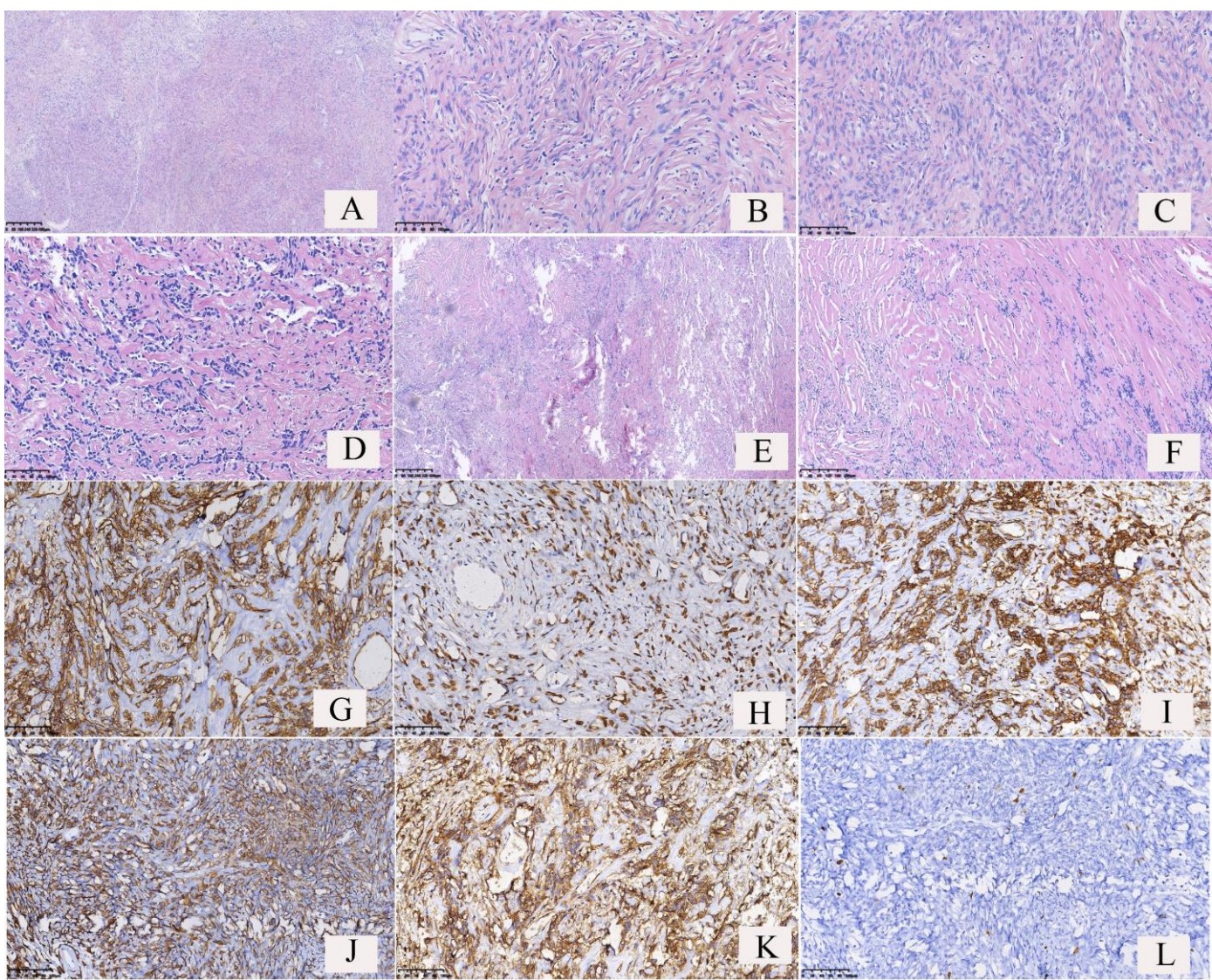

**Figure 4.** Pathological features and immunophenotypes of borderline solitary fibrous tumors (SFTs) of the lung. (**A**) The distribution of the tumor cells showed dense areas and sparse areas, and dilated blood vessels could be found. (**B**) The tumor cells were found in a whorled pattern in dense areas. (**C**) The tumor cells were spindle-shaped and uniform in size. Some tumor cells showed mild atypia. (**D**) Glass-like plastic collagen fibers were interspersed between the tumor cells. The number of spindle tumor cells and collagen fibers was almost equal. (**E**) Hemangiopericytoma-like conformation. (**F**) Numerous collagen fibers were observed in the tumor, but spindle tumor cells were relative few. (**G**) The tumor cells and vascular endothelial cells showed cytomembrane strong positivity for CD34. (**H**) The tumor cells showed nuclear strong positivity for STAT6. (**I**) The tumor cells showed cytoplasmic strong positivity for Vimentin. (**J**) The tumor cells showed both nuclear and cytoplasmic strong positivity for BCL2. (**K**) The tumor cells showed both nuclear and cytoplasmic strong positivity for CD99. (**L**) The number of Ki-67-positive cells was <5% (Ki-67 index <5%).

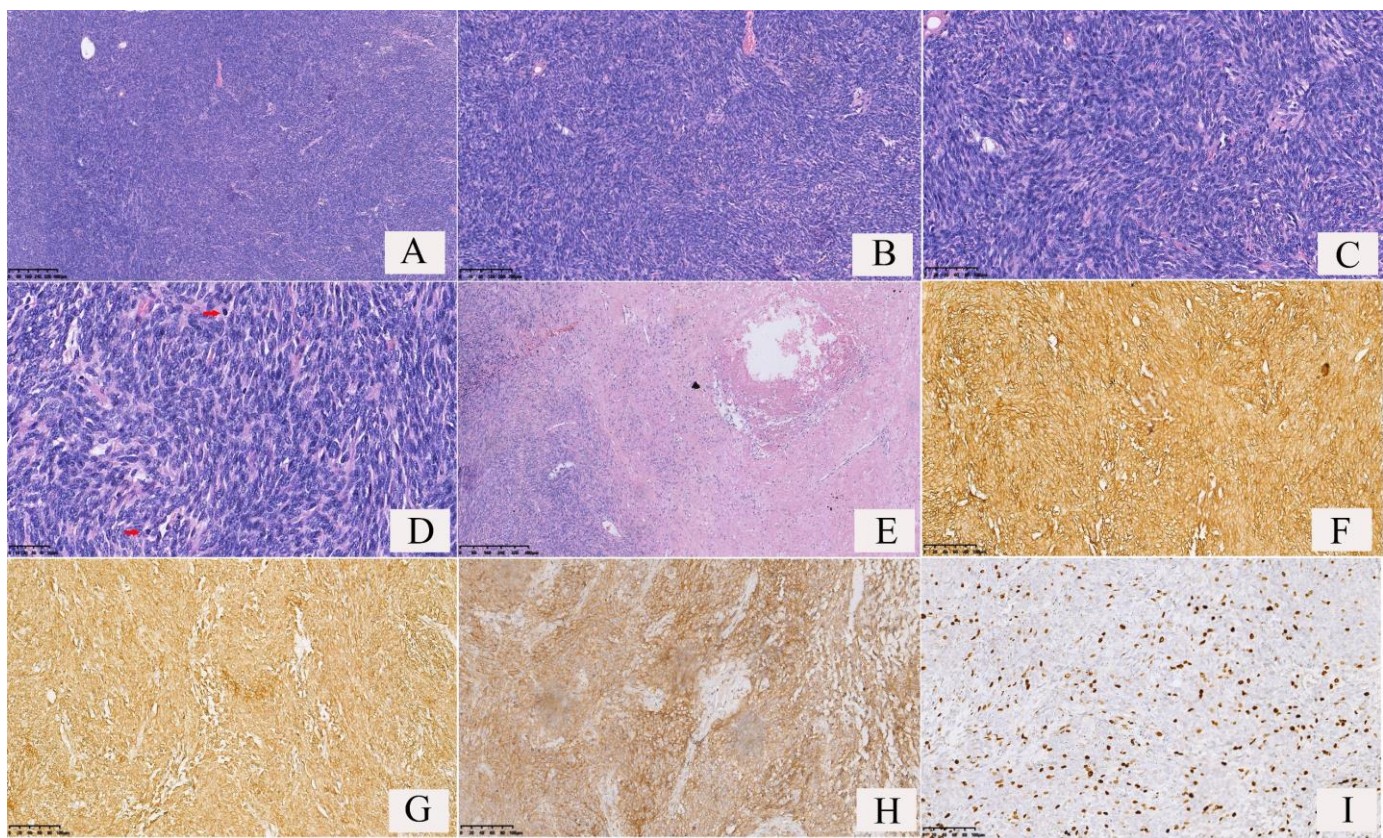

**Figure 5.** Pathological features and immunophenotypes of a malignant solitary fibrous tumor (SFT) of the lung. (**A**) The tumor cells were densely packed, and dilated vessels could be found; (**B**) the tumor cells were distributed around vessels; (**C**) the tumor cells were spindle-shaped with significant atypia; (**D**) frequent mitotic figures were observed (red arrows); (**E**) tumor necrosis was observed; (**F**) the tumor cells and vascular endothelial cells showed cytomembrane strong positivity for CD34; (**G**) the tumor cells showed both cytoplasm and cytomembrane strong positivity for BCL2; (**H**) the tumor cells showed both cytoplasm and cytomembrane strong positivity for CD99; (**I**) the number of Ki-67 positive cells was >20%(Ki-67 index >20%).

**Table 1.** Immunophenotypes of solitary fibrous tumors (SFTs) of the lung.

| Immunohistochemical Markers | CD34 | STAT-6 | BCL-2 | CD99 | Vimentin | PR | ER | Ki-67 (<5%) |
|---|---|---|---|---|---|---|---|---|
| Number of positive cases | 49 | 40 | 40 | 42 | 52 | 6 | 2 | 51 |
| Number of tested cases | 52 | 40 | 51 | 49 | 52 | 9 | 3 | 52 |
| Positive rate | 94.23% | 100% | 78.43% | 85.71% | 100% | 66.67% | 66.67% | 98.07% |

*3.5. Metastatic Risk Assessment*

Age score: 30 cases scored 0 and 22 cases scored 1. Tumor size score: 36 cases scored 0, 11 cases scored 1, 3 cases scored 2, and 2 cases scored 3. Mitotic activity score: 43 cases scored 0, 8 cases scored 1, 1 case (the malignant case) scored 2. Tumor necrosis score: All 52 cases scored 0. Total score: 49 cases scored 0–3, 3 cases scored 4–5, and no case scored 6–7. According to the four-variable risk-stratification model, 49 cases are classified into the low-risk group and 3 cases in the intermediate-risk group.

**4. Discussion**

Solitary fibrotic tumors are fibroblastic tumors characterized by a prominent branching staghorn vasculature and were originally named hemangiopericytomas [6]. Solitary fibrous

tumors can occur in patients of all ages, most commonly in middle-aged people (41–60 years old), with no obvious difference in gender. Solitary fibrous tumors can occur in any part of the body, with 40% occurring in subcutaneous soft tissue, and SFTs are also found in the limbs, head and neck, chest wall, mediastinum, pericardium, retroperitoneum, and abdominal cavity. SFTs occur more commonly in the pleura but are rare in the lung. Of the SFTs occurring in the lung, most occur in the lower lobe of the left lung.

Early in SFT development, patients often have no obvious symptoms. SFTs are often accidentally found by thin-layer CT of the chest during physical examinations. As tumors develop, cough, chest tightness, chest pain, shortness of breath, and other symptoms appear. CT often shows regular and well-demarcated soft tissue density mass, which are nodular, sometimes peripherally lobulated. However, imaging manifestations of SFTs lack specificity and heterogeneity is possible; sometimes, SFTs may be misdiagnosed as other tumors [7]. The radiographic findings reported in this study are consistent with this report. The tumor size of most SFTs ranges from 1 to 10 cm. The maximum diameter of SFTs of the lung in this study varied from 0.7 to 23 cm; most cases of SFTs were in the range of 1–10 cm, accounting for 88.46% (46/52), and only two cases of SFTs were less than 1 cm, with four cases of SFTs over 10 cm.

The conventional pathological-diagnostic method for SFT is a combination of morphological analysis by HE staining and immunophenotypic analysis by immunohistochemical staining. Microscopically, SFTs are variably cellular and composed of cells with oval to spindle-shaped nuclei with minimal cytoplasm and intervening collagen bands arranged in a patternless distribution with areas highly rich in tumor cells while other areas are more hypocellular with a higher percentage of stromal collagen [8]. The morphological characteristics of SFTs of the lung in the present study are consistent with the above manifestations, with tumor cells showing a spindle shape and being relatively uniform, with or without obvious atypia, and separated by glassy collagen fibers. Some patients' SFTs showed a hemangiopericytoma-like conformation. Epithelioid cells with a pathologically mitotic phase and antler-like blood vessels could be found in areas highly rich in tumor cells. CD34, BCL-2, and CD99 are often used as immunohistochemical markers for the differential diagnosis of SFTs, but these markers are not specific. The specific, sensitive, and more recommended marker is STAT 6. In this study, of the 52 SFTs tested for CD34 expression, 49 were positive; of the 51 tested for BCL-2 expression, 40 were positive; of the 49 tested for CD99 expression, 42 were positive; and of the 40 tested for STAT 6 expression, all were positive, with a 100% positive rate. Of the fifty-two SFTs tested for Vimentin expression, all were positive, with a 100% positive rate. Some SFTs showed ER and PR expressions. The results showed that STAT 6 and Vimentin expression have the strongest specificity and sensitivity for SFTs. The immunostains of CD34, STAT6 and Ki-67 are typically pleural based and show involvement of lung parenchyma; radiologic findings help to find the location of SFTs. The chest CT findings of the 52 cases supported the lung location of SFTs. SFTs are mostly borderline and occasionally malignant. However, the expression of the immunohistochemical markers CD34, BCL-2, CD99, and STAT 6 did not differ between the borderline and malignant solitary fibrous tumors and therefore could not be used in distinguishing benign and malignant SFTs. The main criteria for classifying the malignant variants of SFT of the lung and pleura were described by England et al., Vallat-Decouvelaere et al. and Krsková et al. [9–11]. They established the following features suggestive of malignancy: the presence of a high density of tumor cells, high cellularity and mitotic activity (more than 4 mitotic figures per 10 high-power fields), significant atypia, pleomorphism, hemorrhage, presence of necrosis, stromal or vascular invasion, size exceeding 10 cm, and a significant increase in the Ki-67 index. Ki-67/MIB-1 proliferation index >10% is generally considered borderline or low-grade malignant. Of the 52 SFTs of the lung in this study, 51 were borderline tumors, with a Ki-67 $\leq$ 5% and only one SFT was malignant, with a Ki-67 > 20%. In the one case of malignant SFT, microscopically, the tumor cells were spindle-shaped and densely arranged, dilated blood vessels were visible, tumor cells were distributed around blood vessels, and obvious atypia, necrosis (7%), and

multiple mitotic figures (>4 mitotic figures per 10 high-power fields) were found; all of which met the characteristics of malignant SFTs.

The histopathology of SFTs is variable but typically demonstrates either fibroblastic or cellular spindle-cell morphology [12]. SFTs must be differentiated from other spindle-cell tumors, such as fibrosarcomas (strip/fish bone-like arrangement, with obvious atypia), leiomyosarcomas (spindle cells are blunt at both ends, positive expression of Desmin and SMA), schwannomas (alternating areas Antoni A and Antoni B; positive expression of SOX-10 and S-100), synovial sarcomas (interstitial and epithelial differentiation), and thymomas (strongly positive expression of pan-CK and P63). Both microscopic morphological and immunohistochemical results in this study excluded other spindle cell tumors.

In recent years, studies have shown that almost all SFT patients have a NAB2-STAT6 gene fusion, which provides a new basis for difficult-to-diagnose cases of SFTs [13]. Patients were divided into two groups on the basis of the NAB2-STAT6 gene fusion, namely, the STAT6-TAD and STAT6-full groups, and patients in the STAT6-TAD group had a higher risk of relapse, which has important implications for SFT prognosis judgment [14]. The patients in the present study were clearly diagnosed by pathologically morphological features and immunohistochemical results; therefore, the NAB2-STAT6 gene-fusion test was not performed. Resection with free margins is considered to be the treatment for SFT located either in the lung or on the pleura. Most SFT patients with borderline tumors have slow tumor growth and a good prognosis, but a few SFTs can relapse and need early and complete resection and long-term follow-up [7,15,16]. Whether the tumor has a vascular pedicle may influence its staging and prognosis [17,18]. Very few SFTs are malignant and show common recurrence, visible metastasis, and lead to patient death. Demicco reported a novel risk-stratification scheme for solitary fibrous tumors incorporating patient age, tumor size, tumor necrosis, and mitotic activity to predict the risk of metastasis [5]. According to the risk-stratification scheme, in the 52 cases of SFTs of this study, 49 cases are classified into the low-risk group and 3 cases in the intermediate-risk group, with no cases in the high-risk group.

In this study, 6 patients were lost to follow-up, and 46 patients were followed up from 8 months to 13 years and 6 months, CT re-examination was carried out every year. Among the 46 patients, only 1 patient of SFT combined with squamous cell carcinoma died one year after surgical resection. As the size of squamous cell carcinoma was bigger (8.5 cm × 6.0 cm × 5.0 cm) and the age of the patient was 70 years, we inferred that the patient died of squamous cell carcinoma, not of solitary fibrous tumor. The prognosis for the other 45 patients, including 1 patient with malignant SFT, 1 patient combined with microinvasive adenocarcinoma, and 1 patient with the largest SFT (23 cm×20 cm×9 cm), was good, and no recurrence or metastasis was found. However, long-term follow-up is needed especially for the three cases in intermediate risk.

## 5. Conclusions

In summary, solitary fibrous tumors (SFTs) of the lung occur more frequently in middle-aged and elderly people, and there is no significant difference in SFT incidence between men and women. SFTs of the lung most frequently occur in the lower lobe of the left lung. The tumor size is mostly between 1 and 10 cm. Chest CT often shows regular and well-demarcated soft tissue density mass. Pathologically, SFTs of the lung may be borderline or malignant, similar to SFTs occurring at other sites. Immunohistochemistry shows the positive expression of CD34, STAT6, Vimentin, BCL2, and CD99. Malignant SFTs are very rare and share the same immunophenotype and molecular features as borderline SFTs. Our findings provide further information for understanding the clinical manifestations, radiologic features, pathological features, and immunophenotype of SFTs of the lung, and may improve the awareness of this tumor for clinicians, radiologists, and pathologists.

*Strengths and Limitations of this Study*

Solitary fibrous tumors (SFTs) can occur in all organs of the body but rarely occur in the lung. The current reports of SFTs occurring in the lung are fewer than 10 cases; in this study, 52 cases of SFTs occurring in the lung were reported and the clinical, radiological, and morphological features, immunophenotypes, and prognosis of SFTs of the lung were summarized. However, the NAB2-STAT6 gene fusion test was not performed on patients in this study.

**Author Contributions:** Y.-X.W. collected and analyzed the data, and wrote the original manuscript. Y.Z., S.-S.F., Y.-S.Z. and X.-R.P. collected and analyzed the data. X.Z. designed the project and revised the manuscript. All authors have read and agreed to the published version of the manuscript.

**Funding:** This study was supported by grants from the Joint Projects of Applied Basic Research of Kunming Medical University and Yunnan Provincial Department of Science and Technology (No. 202101AY070001-010; No. 2019FE001-220).

**Institutional Review Board Statement:** The study was conducted according to the guidelines of the Declaration of Helsinki, and approved by the Ethical Review Board of the First Affiliated Hospital of Kunming Medical University (protocol code: 2022L023, the date of approval: 16 September 2022.).

**Informed Consent Statement:** Patient consent was waived due to our retrospective evaluation of the patients' stored samples that would not involve more than minimal risk to the subjects. Moreover, we were unable to obtain consent from study subjects because patients were initially treated in the past, and it is not practical to locate and obtain consent at this time.

**Data Availability Statement:** The data presented in this study is available in this article.

**Conflicts of Interest:** The authors declare no conflict of interest.

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
