# Peer review of "Solitary Fibrous Tumors of the Lung: A Clinicopathological Analysis of 52 Cases"

_curroncol, doi:10.3390/curroncol30020138_

Round 1
Reviewer 1 Report
This paper is about clinicopathological features of pulmonary solitary fibrous tumor (SFT).
This is a large group yet not updated or new. A few updates need to be addressed. these are substratified by risk (Demicco paper and WHO) and usually are at least borderline to low grade when deep, older patient, large, and high grade is exceedingly rare. the parameters of mitoses and necrosis are not discussed. >10% MIB is generally considered borderline or low grade malignant. The only immunostains that are relevant are CD34, and STAT6, and Ki-67 These are typically pleural based and involvement of lung parenchyma and more correlation with radiologic findings would be of interest.
Author Response
Comments: This paper is about clinicopathological features of pulmonary solitary fibrous tumor (SFT). This is a large group yet not updated or new.
A few updates need to be addressed. these are substratified by risk (Demicco paper and WHO) and usually are at least borderline to low grade when deep, older patient, large, and high grade is exceedingly rare. the parameters of mitoses and necrosis are not discussed.
Response: We are very grateful for the reviewer’s precious comments and suggestions,which are very helpful for improving our manuscript.
We have cited Demicco paper”Demicco, E., Wagner, M., Maki, R. et al. Risk assessment in solitary fibrous tumors: validation and refinement of a risk stratification model. Mod Pathol 30, 1433–1442 (2017). https://doi.org/10.1038/modpathol.2017.54.” And made a metastatic risk assessment on the 52 cases in this study(in introduction,methods,results and discussion).The risk factors, mitoses and necrosis,age and tumor size are scored according to the risk stratification model by Demicco.
Comments:>10% MIB is generally considered borderline or low grade malignant.
Response:A significant increase of Ki-67/MIB-1 is one of indicators suggestive of malignancy. In this study,51 were borderline, with a Ki-67/MIB-1≤ 5%,and 1 case was malignant, with a Ki-67 /MIB-1> 20%. The diagnosis of borderline or malignant SFTs is made by multiple indicators including the density of tumor cells, mitotic activity, atypia, tumor size, necrosis, stromal or vascular invasion, and Ki-67/MIB.
Comments:The only immunostains that are relevant are CD34, and STAT6 and Ki-67 These are typically pleural based and involvement of lung parenchyma and more correlation with radiologic findings would be of interest.
Response:We added“The immunostains of CD34, and STAT6 and Ki-67 are typically pleural based and involvement of lung parenchyma, radiologic findings help to find the location of SFTs. The chest CT findings of 52 cases support the lung location of SFTs.”in discussion of the manuscript.
Reviewer 2 Report
This is a retrospective review of 52 cases of solitary fibrous tumor with primary origin in the lung. Overall, the authors have presented the data very well. Following points to be noted:
#1 is data on Next Generation sequencing available for the borderline or malignant SFTs? If yes then it would be helpful to include
#2 one of the most accepted risk ratification criteria for SFT has been validated by Demicco et al. Demicco, E., Wagner, M., Maki, R. et al. Risk assessment in solitary fibrous tumors: validation and refinement of a risk stratification model. Mod Pathol 30, 1433–1442 (2017). https://doi.org/10.1038/modpathol.2017.54. How do these 52 cases fit with the restratification as defined in this article. This reference article should be cited and discussed in the manner
#3 was a PET scan done on any of these patients. Were then noted to be avid on the PET scan? Is there a difference in uptake between borderline and malignant case?
#4 did all patients undergo full staging including abdomen and pelvis CT scan as part of the original diagnosis. If yes, then it should be included in the method section
Author Response
Comments:This is a retrospective review of 52 cases of solitary fibrous tumor with primary origin in the lung. Overall, the authors have presented the data very well. Following points to be noted:
Response: We are very grateful for the reviewer’s precious comments and suggestions,which are very helpful for improving our manuscript.
Comments:#1 is data on Next Generation sequencing available for the borderline or malignant SFTs? If yes then it would be helpful to include.
Response:Sorry we didn’t do Next Generation sequencing for the 52 cases of SFTs. And it is unclear whether the data on Next Generation sequencing have difference between borderline and malignant SFTs. The patients in the present study were clearly diagnosed by pathologically morphological features and immunohistochemical results.
Comments:#2 one of the most accepted risk ratification criteria for SFT has been validated by Demicco et al. Demicco, E., Wagner, M., Maki, R. et al. Risk assessment in solitary fibrous tumors: validation and refinement of a risk stratification model. Mod Pathol 30, 1433–1442 (2017). https://doi.org/10.1038/modpathol.2017.54. How do these 52 cases fit with the as defined in this article. This reference article should be cited and discussed in the manner
Response:We have cited this reference article and discussed how do these 52 cases fit with the risk ratification criteria for SFT validated by this reference article. We made a metastatic risk assessment on the 52 cases in this study(in introduction,methods,results and discussion).The risk factors, mitoses and necrosis,age and tumor size are scored according to the risk stratification model by Demicco.
Comments: #3 was a PET scan done on any of these patients. Were then noted to be avid on the PET scan? Is there a difference in uptake between borderline and malignant case?
Response: None of these patients received a PET scan as it is not included in the coverage of China Medical Insurance and must be paid by individuals.
Comments:#4 did all patients undergo full staging including abdomen and pelvis CT scan as part of the original diagnosis. If yes, then it should be included in the method section.
Response:Not all patients underwent full staging CT scan as part of the original diagnosis. Part of the patients underwent full staging CT scan including abdomen and pelvis as part of the original diagnosis, but no abnormality was found in abdomen and pelvis or other parts except lung.
Round 2
Reviewer 1 Report
Prognostic information added. It